# Self-Supervised Learning Through Efference Copies

**Franz Scherr**[1]*
Huawei Technologies
franz.scherr@huawei.com

**Qinghai Guo**[2]
Huawei Technologies
guoqinghai@huawei.com

**Timoleon Moraitis**[1]*
Huawei Technologies
timoleon.moraitis@huawei.com

## Abstract

Self-supervised learning (SSL) methods aim to exploit the abundance of unlabelled data for machine learning (ML), however the underlying principles are often method-specific. An SSL framework derived from biological first principles of embodied learning could unify the various SSL methods, help elucidate learning in the brain, and possibly improve ML. SSL commonly transforms each training datapoint into a pair of views, uses the knowledge of this pairing as a positive (i.e. non-contrastive) self-supervisory sign, and potentially opposes it to unrelated, (i.e. contrastive) negative examples. Here, we show that this type of self-supervision is an incomplete implementation of a concept from neuroscience, the Efference Copy (EC). Specifically, the brain also transforms the environment through efference, i.e. motor commands, however it sends to itself an EC of the full commands, i.e. more than a mere SSL sign. In addition, its action representations are likely egocentric. From such a principled foundation we formally recover and extend SSL methods such as SimCLR, BYOL, and ReLIC under a common theoretical framework, i.e. Self-supervision Through Efference Copies (S-TEC). Empirically, S-TEC restructures meaningfully the within- and between-class representations. This manifests as improvement in recent strong SSL baselines in image classification, segmentation, object detection, and in audio. These results hypothesize a testable positive influence from the brain's motor outputs onto its sensory representations.

## 1 Introduction

Deep Learning (DL) has drawn inspiration from Neuroscience and also offers models for understanding aspects of the brain (Richards et al., 2019). DL has been extremely successful, largely owing to labelled big datasets. However, such labelling is a costly procedure carried out by human supervisors. Fully unsupervised ML techniques do exist, however they rarely reach the performance of supervised learning (Moraitis et al., 2022; Journé et al., 2022). On the other hand, recently, a category of algorithms that are self-supervised has emerged. In self-supervised learning (SSL), the model itself generates the supervisory signal, so that human supervision is not needed, and then uses that signal for supervised learning. Recent SSL algorithms generate the supervisory signal by using pairs of inputs where it is known whether they are of the same or of a different instance, therefore self-generating positive or negative labels. Examples can be associated as being positive, e.g. based on their temporal proximity, if the input is in a sequential domain (Oord et al., 2018). Advanced SSL algorithms generate themselves the positive pairs of inputs, by augmenting the training dataset (He et al., 2020; Chen et al., 2020a; Grill et al., 2020; Caron et al., 2020; Mitrovic et al., 2021). Further improving these algorithms even has the potential to outperform supervised learning (Tomasev et al., 2022), because more information exists in the comparison of complete input pairs than in individual human-labelled examples. Therefore, improving SSL consists in devising representation-learning methods that better capture that information. However, conceptual frameworks that unify these

---

[1]Huawei Zurich Research Center, Switzerland, [2]Huawei ACS Lab, Shenzhen, China, *Corresponding author

36th Conference on Neural Information Processing Systems (NeurIPS 2022).

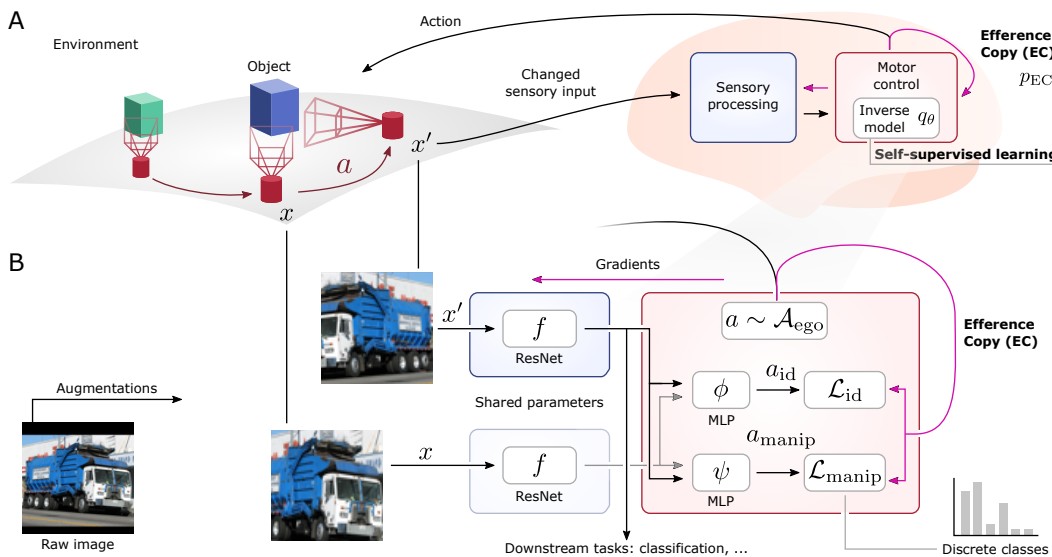

Figure 1: **Efference copy (EC). A**) Sensory-motor system: Efferent actions $a$ (change in focus, manipulations, etc.) yield changed sensory input $x'$. The internal copy of the motor command, i.e. the EC, we propose, may be used as a self-supervisory signal for learning an inverse model $q_\theta$. **B**) Abstract ML setting: The action space comprises switching and augmenting images. Sensory processing and inverse models are implemented as neural networks, trained through EC's feedback.

principles of existing SSL methods, and guide towards new improved ones, are scarce (Balestriero and LeCun, 2022).

## 2 Efference copies in the central nervous system

The operation of the biological central nervous system (CNS), which appears to learn mostly without external supervision, may provide such a framework. Conversely, ML simulations within such a framework may also generate testable hypotheses for biological SSL. In the present study we take this abstract hope and formulate it as a concrete link from SSL to a specific mechanism in the CNS. We begin by observing first, that the CNS of vertebrate animals is believed to have evolved with the main purpose of performing sensory-motor control and learning, and second, that the data manipulations that augment the training examples in ML implementations of SSL can be viewed as motor actions. The search for analogies then can focus on looking for possible self-supervisory signals within biological motor control and learning.

A particularly well-suited and well-established signal in the sensory-motor system is that of the Efference Copy (EC) (von Helmholtz, 1867; McNamee and Wolpert, 2019) or Corollary Discharge (Sperry, 1950). Namely, it has been shown that when a component of the CNS addresses the body's muscles with an efferent, i.e. outgoing, motor command or action, often it also sends a copy to the CNS itself, see Fig. 1A. ECs have multiple functions and abundant supporting evidence (Kennedy et al., 2014; McNamee and Wolpert, 2019; Kilteni et al., 2020; Latash, 2021). For example, certain motor commands responsible for the locomotion of frogs are generated in the spinal cord, but are copied to the brainstem, which is responsible for motor control of the eye (von Uckermann et al., 2013). The body-movement-related disturbances to the visual field are then predicted and appropriately counteracted by eye movements that stabilize the frog's gaze. Therefore, one function of EC is to coordinate different motor controllers of the body. Another function of EC is to focus sensory processing on externally-generated and unpredicted stimuli by cancelling predictable sensations of self-generated actions. E.g., humans cannot tickle themselves effectively, because by using its ECs the CNS predicts the sensory consequence of its own action, and cancels it before it is perceived (Blakemore et al., 1998). The role of ECs in humans is actually broader and very central to motor control. Specifically, the control of bodily movements involves forward internal models that the brain maintains, i.e. models that predict the sensory inputs that result from each motor command (Kawato, 1999; McNamee and Wolpert, 2019). These forward models rely on access to

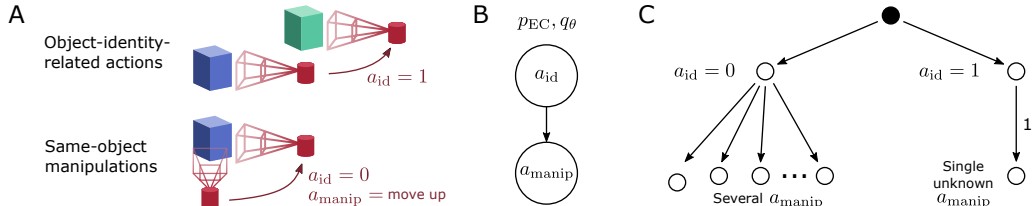

Figure 2: **Actions. A**) Categories: New objects can be brought into focus $a_{\mathrm{id}} = 1$ (e.g. saccades). Else ($a_{\mathrm{id}} = 0$), the same object can be manipulated by $a_{\mathrm{manip}}$ (e.g. moving). **B**) Dependency structure as a graphical model. **C**) Corresponding decision tree if $a_{\mathrm{manip}}$ is further assumed to be discrete.

motor commands to generate their predictions, and that access is provided by ECs. Importantly, motor control also involves inverse models, which map representations of targeted movement sensations to their possible actions (Rizzolatti et al., 1998; Kawato, 1999) (Fig. 1A). In addition to motor control, ECs also underlie motor *learning* (Witney et al., 1999; Troyer and Doupe, 2000; Diedrichsen et al., 2003; Engert, 2013; Brownstone et al., 2015). For example, when learning an inverse model, the structures that calculate errors must access the efference.

Given the pervasive role of ECs in sensory processing, motor control, and motor learning, we hypothesize that ECs could play a key role in the learning of sensory representations too, and that it does so through the learning of inverse models. More specifically, we hypothesize that, if the EC acts as a self-supervisory learning signal, then it improves the sensory learning process, e.g. improving the later classification of input examples. Rather than physiological experiments, or biologically detailed simulations, we will test the hypothesis in an abstract ML setting. Nevertheless, we will use mechanisms that do have plausible biophysical implementations. In addition, our model could improve ML methods by providing more of the information content of paired input datapoints to SSL. That is because ECs can be rich and diverse signals, i.e. they can provide the full description of the actions that generate input pairs, and can do so for varied types of actions.

## 3 S-TEC: Self-supervision Through Efference Copies

### 3.1 Definitions and key principles

Our basic assumption is that *(a) an EC is available*, i.e. a copy of the efferent motor commands, or actions. The essence of our strategy is to use this EC as a target label to *(b) learn an inverse model*, mapping sensory inputs to the motor outputs that caused the changed inputs in the first place (Kawato, 1999). We assume *(c) a hierarchical model*, e.g. a multilayer neural network. We conjecture that a model that improves on this motor-oriented task, will also improve its intermediate sensory representations as a direct consequence, which are then useful to a wider variety of sensory tasks. In our experiments, we use the representation for classification. In order to concretize the model, let $x \in \mathcal{D}$ generally denote sensory inputs. Furthermore, we define the motor commands as actions $a \in \mathcal{A}_{\mathrm{ego}}$ that result in transformed inputs $x' = T(x, a)$, denoting with $T$ the transformation function, see also Fig. 1A. In the following, we will simply write the EC as a probability distribution $p_{\mathrm{EC}}(a|x, x')$ to indicate the distribution of values it will assume given the sensory inputs are $x$ before an action was taken, and are $x'$ thereafter. We utilize this as a ground truth that the inverse model needs to predict. More formally, we denote the to-be-learned inverse sensory-motor mapping by $q_\theta(a|x, x')$ with free parameters $\theta$ that surmise synaptic weights. Learning then is the minimization of the discrepancy between the ground truth $p_{\mathrm{EC}}$ and our model $q_\theta$:

$$\min_\theta \underset{\substack{x, \tilde{a} \in \mathcal{D} \times \mathcal{A}_{\mathrm{ego}} \\ x' = T(x, \tilde{a})}}{\mathbb{E}} \left[ \underbrace{D_{\mathrm{KL}}(p_{\mathrm{EC}}(a|x, x'); q_\theta(a|x, x'))}_{=:\mathcal{L} \ (\mathrm{Loss})} \right]. \tag{1}$$

In the above formulation, we denote with $D_{\mathrm{KL}}$ the Kullback-Leibler divergence, and introduce the loss function $\mathcal{L}$ that will be helpful later. The broad concept given in Eq. (1) is so far agnostic to the specific types of actions and sensory inputs. To render the matter more concrete, and to align it with the examples for sensory modalities in Section 2, we will focus on the visual sensory domain. This also facilitates the validation of our approach by ML experiments on contemporary datasets and architectures, see Section 4. We assume that only one type of sensory object is observed with

each sensory input. We denote the set of possible actions as $\mathcal{A}_{\text{ego}}$. To account for the fact that the model concerns sensory-motor control in the physical world, *(d) actions must account for two types of sensory transformations* (see Fig. 2A), namely:

- *(d1) Object-identity-related actions* $a_{\text{id}}$. This category of action switches between sensed objects, e.g. by a saccade of the eyes, bringing entirely new objects into focus, or not. In the context of standard vision datasets that are comprised of static images, we simply exchange the currently viewed image with a randomly sampled new one. The two types of actions in this category, i.e. switching or not, are $a_{\text{id}} = 1$ or $a_{\text{id}} = 0$ respectively.

- *(d2) Same-object manipulations* $a_{\text{manip}}$. This category is identity-preserving, i.e. maintaining the sensed object but the observer actively manipulates it or its view, e.g.: turning the object, or moving to a closer vantage point. With static images, this kind of transformation is naturally formed by commonly used image augmentation operations. E.g. spatial transformations that crop an image with random size and random aspect ratio can simulate the movement to a different point at a closer distance whereas mirroring the image horizontally corresponds well to rotating a symmetric 3D object, see Fig. 1B. We denote an action that transforms one augmented view $x$ into the other augmented view $x'$ by $a_{\text{manip}}$.

The object-identity-related actions are of two types $a_{\text{id}} \in \{0, 1\}$. Therefore, this part of the action representation is categorical. Based on this, *(e) we model the entire action representation as categorical*, i.e. including $a_{\text{manip}}$. This is to follow the biological evidence that the brain maintains uniform principles throughout its organization, e.g. throughout the cortex (Douglas et al., 1989), including motor areas (Bastos et al., 2012). Moreover, there is significant evidence that this uniform organization does specifically have a categorical structure, were different actions are represented by different clusters of neurons (Graziano, 2016). Importantly, this allows learning the associated inverse model by means of a classification task, as will be introduced later.

As the overall action $a$ is composed by two parts, i.e. $a = (a_{\text{id}}, a_{\text{manip}})$, we can summarize this categorical structure as a graphical model and decision tree, shown in Fig. 2B and C. Hence, given the same object continues to be in focus, i.e. $a_{\text{id}} = 0$, then there exist several options for the object-manipulating action $a_{\text{manip}}$. In the other case, where focus is switched to a different object, i.e. $a_{\text{id}} = 1$, there is no value of the object-manipulation action $a_{\text{manip}}$ that relates $x$ and $x'$. To formally represent this in the decision tree, we assign all probability to some unknown $a_{\text{manip}}$ in that case.

So far we have not described how each class of action $a_{\text{manip}}$, and therefore its copy EC, is parametrized by the motor controller. Based on the fact that EC conveys to the observer the action that himself is taking, it is appropriate to *(f) use an egocentric representation of actions* instead of aligning the actions with an allocentric reference point, i.e. with the environment. To do so, notice that the spatial transformations introduced in (d2) are affine, thus can be represented with their associated transformation matrices. This logical parametrization allows us to conveniently compute the egocentric action that is needed to turn $x$ into $x'$: By multiplication of the transformation matrix that gave rise to one view from the original with the inverted transformation matrix that gave rise to the other. This highlights a difference to the allocentric representation of actions that was chosen in other work (Lee et al., 2021), where transformations were aligned to the original, allocentric reference frame (i.e. differences of scales rather than their quotient as it would emerge here).

## 3.2 Formalism

Through the preceding dependency structure (Fig. 2B, C), the inverse model naturally decomposes into two more specific inverse models, where one is attributed to object-identity-related actions, and the other to the same-object manipulations, to which we simply refer to as "**identity-related inverse model**" $q_\theta(a_{\text{id}}|x, x')$ and "**manipulation-related inverse model**" $q_\theta(a_{\text{manip}}|a_{\text{id}}, x, x')$ respectively. Therefore we have that $q_\theta(a|x, x') = q_\theta(a_{\text{id}}|x, x')q_\theta(a_{\text{manip}}|a_{\text{id}}, x, x')$. Applying the same also for the ground truth $p_{\text{EC}}$ enables us to split the loss function into separate parts $\mathcal{L} = \mathcal{L}_{\text{id}} + \mathcal{L}_{\text{manip}}$ that reflect learning of the identity-related inverse model and learning of the manipulation-related inverse model correspondingly. More precisely, the loss dedicated to the identity-related inverse model is given by $\mathcal{L}_{\text{id}} = D_{\text{KL}}(p_{\text{EC}}(a_{\text{id}}|x, x'); q_\theta(a_{\text{id}}|x, x'))$, while similarly, the loss dedicated to the manipulation-related inverse model is given by $\mathcal{L}_{\text{manip}} = D_{\text{KL}}(p_{\text{EC}}(a_{\text{manip}}|a_{\text{id}}, x, x'); q_\theta(a_{\text{manip}}|a_{\text{id}}, x, x'))$, see also Appendix D for details.

In practice, we also include regularization losses $\mathcal{L}_{\text{reg}}$, see Appendix C, and weight the relative importance of the loss terms by hyperparameters $\lambda$. Therefore, the loss that we consider is given by:

$$\mathcal{L} = \mathcal{L}_{\text{id}} + \lambda_{\text{manip}}\mathcal{L}_{\text{manip}} + \lambda_{\text{reg}}\mathcal{L}_{\text{reg}} . \tag{2}$$

**Instantiation of the inverse models.** The specific formulation of $q_\theta(a|x, x')$ determines the loss function that will be optimized. We consider several options for the formulation of $q_\theta(a_{\text{id}}|x, x')$ that allow us to recover various contemporary approaches for SSL as we discuss in Results Section 4.1. On the other hand, for the manipulation-related inverse model, we opted for a categorical action representation that clusters similar actions as advocated in key principle (e). We implemented this by subdividing the support of each single component $a_{k,\text{manip}}$ of $a_{\text{manip}}$ (consisting of 6 components for the affine transformation) into a number of $K$ discrete bins. We indicate these discretized versions of the real actions with a hat $\widehat{\cdot}$ and define the probability of $a_{k,\text{manip}}$ being in bin $b$ as:

$$q_\theta(\widehat{a}_{k,\text{manip}} = b|a_{\text{id}} = 0, x, x') = \frac{\exp\left(\psi_{k,b}(f(x), f(x'))\right)}{\sum_j \exp\left(\psi_{k,j}(f(x), f(x'))\right)} , \tag{3}$$

where we have introduced a feature extractor $f$ (ResNets in our case, see Fig. 1) and the functions $\psi_{k,j}$ (for which we used MLPs) to express the model's belief that $a_{k,\text{manip}}$ assumes a value in discrete bin $j$. Note that the functions $f$ and $\psi$ both are learnable, but the dependence on $\theta$ is omitted for brevity. We refer to Fig. 3 for an ablation study on alternative instantiations of the manipulation-related inverse model.

## 4 Results

### 4.1 Recovering contrastive & non-contrastive SSL from the identity-related inverse model

Depending on the specific instantiation of the identity-related inverse model, we recover several common approaches for SSL using the concept of ECs. In particular, we show that based on the choice of the learned $q_\theta(a_{\text{id}}|x, x')$, we can recover from the identity-related loss either contrastive losses (i.e. instance discrimination) such as employed in SimCLR (Chen et al., 2020a), ReLIC (Mitrovic et al., 2021) or ReLICv2 (Tomasev et al., 2022), or non-contrastive approaches such as BYOL (Grill et al., 2020) (see below, and Appendices D.7.1, D.7.2, and D.7.3).

First, we consider here as an example the identity-related inverse model that gives rise to the contrastive loss of SimCLR (Chen et al., 2020a). We define this inverse model's probability of no identity-switch, i.e. $a_{\text{id}} = 0$, in the common way used for the positive view in contrastive learning (Chen et al., 2020a), for which we adopt the notation provided by Mitrovic et al. (2021):

$$q_\theta(a_{\text{id}} = 0|x, x') = \frac{\exp\left(\phi(f(x), f(x'))/\tau\right)}{\sum_{x_n \in \{x'\} \cup C} \exp\left(\phi(f(x), f(x_n))/\tau\right)} . \tag{4}$$

Here, $\phi$ is a function that computes a similarity between the features produced by $f$, i.e. it compares the intermediate sensory representations. We defined it as a scalar product between projected features: $\phi(h, h') := \sum_i g_i(h)g_i(h')$, whereas $g$ is a multi-layer perceptron (MLP), following typical choices in the literature (see also Appendix B). The scalar $\tau$ is a temperature hyperparameter, and the set $C$ is composed of additional candidate inputs to which $x$ is compared to (through the denominator). Note that $g$ is learnable also, thus depending on $\theta$.

We assume that the EC is a perfect copy of $a$, hence $p_{\text{EC}}(a|x, x')$ assigns all probability to the true action $a$ that was applied. In doing so, we obtain an upper bound of the objective (2) that we use for our S-TEC experiments (see Appendix D for the derivation). Its associated component dedicated to the identity-related inverse model is the typical contrastive learning objective:

$$\mathcal{L} \leq -\log \frac{\exp(\phi(f(x), f(x''))/\tau)}{\sum_{x_n \in \{x'\} \cup C} \exp\left(\phi(f(x), f(x_n))/\tau\right)}$$
$$- \lambda_{\text{manip}} \sum_k \log \frac{\exp\left(\psi_{k,j''}(f(x), f(x''))\right)}{\sum_j \exp\left(\psi_{k,j}(f(x), f(x''))\right)} + \lambda_{\text{reg}}\mathcal{L}_{\text{reg}} , \tag{5}$$

where we introduced $x''$ to always represent an input that is related to $x$ through $a_{\text{id}} = 0$. We use $x''$ also for the second term that concerns the manipulation-related inverse model, to reflect that the loss

is only applied in that condition. For additional convenience we write $j''$ to refer to the bin in which the value of $a_{k,\mathrm{manip}}$, relating $x$ and $x''$, falls.

By integrating a confidence-estimate into the identity-related inverse model, we obtain and extend SSL methods such as ReLIC and ReLICv2. Furthermore, by defining the identity-related inverse model as a normal distribution, we obtain non-contrastive losses such as that of BYOL. For these derivations, see Appendices D.7.2 and D.7.3. Interestingly, the loss for VICReg (Bardes et al., 2022) is of the same type as derived from S-TEC's principles for a normally-distributed identity-related inverse model.

## 4.2 Experimental evaluation

Table 1: Accuracies obtained with linear classification (mean and std over 5 independent runs).

| Architecture | Method | CIFAR-10 | CIFAR-100 | STL-10 |
|---|---|---|---|---|
| ResNet-18 | SimCLR repr. (Chen et al., 2020a) | 91.5 ±0.1 | 65.3 ±0.3 | **91.5** ±0.4 |
| | ReLIC repr. (Mitrovic et al., 2021) | 91.5 ±0.2 | 65.6 ±0.3 | 91.4 ±0.2 |
| | MoCo v2 repr. (Chen et al., 2020c) | 90.9 ±0.2 | 65.7 ±0.4 | 88.6 ±0.6 |
| | BYOL repr. (Grill et al., 2020) | 92.0 ±0.2 | 66.4 ±0.4 | 91.1 ±0.2 |
| | S-TEC (ours) | 92.0 ±0.2 | 66.6 ±0.3 | **91.6** ±0.2 |
| | S-TEC*(ours) | **92.6** ±0.1 | **67.6** ±0.4 | 91.4 ±0.2 |
| ResNet-50 | SimCLR repr. (Chen et al., 2020a) | 93.2 ±0.2 | 70.8 ±0.2 | 94.0 ±0.1 |
| | ReLIC repr. (Mitrovic et al., 2021) | 93.2 ±0.2 | 70.8 ±0.2 | 93.8 ±0.1 |
| | S-TEC (ours) | **93.9** ±0.2 | 71.9 ±0.4 | **94.3** ±0.2 |
| | S-TEC*(ours) | **94.0** ±0.1 | **72.4** ±0.1 | 93.9 ±0.2 |

In the preceding Sections we have derived our framework that connects the concept of ECs to current methods for SSL. Here, we present the results of our experimental evaluations that aim to assess the quality of the representations that can be learned with our approach. For this purpose, we considered various image datasets, including CIFAR-10/100 (Krizhevsky et al., 2009), STL-10 (Coates et al., 2011) as well as the ImageNet ILSVRC-2012 dataset (Russakovsky et al., 2015), and compare S-TEC against several other SSL algorithms, such as SimCLR (Chen et al., 2020a), MoCo v2 (Chen et al., 2020c), BYOL (Grill et al., 2020) and ReLIC (Mitrovic et al., 2021). We follow the same procedure for all experiments, where we first performed SSL and subsequently determine the class prediction accuracy of a linear classifier that is trained on the emergent representations. During SSL the same data augmentation methods as in Chen et al. (2020a) are applied throughout, i.e. also colour augmentations, but these were not considered for training an inverse model with S-TEC, see also Appendix A for details to the augmentations applied. We adopted the ResNet v1 framework (He et al., 2016), and used specifically ResNet-18 and ResNet-50 architectures as the feature extractor $f$, while $\phi$ and $\psi$ were generally implemented as multi-layer perceptrons (MLPs), see Fig. 1B, and Appendix B for architectural details. These networks were optimized during SSL by gradient-descent using the adaptive rate scaling of the LARS algorithm (You et al., 2017) with learning rate warmup and decay. If not otherwise stated, SSL was performed for 1,000 epochs. See Appendix C for details to the optimization with SSL and for details of training the linear classifier. For all our comparisons and ablation studies, we stress that the **overlap in the implementation is maximal**. Especially for the comparison between SimCLR and S-TEC (and between ReLIC and S-TEC*), we emphasize that

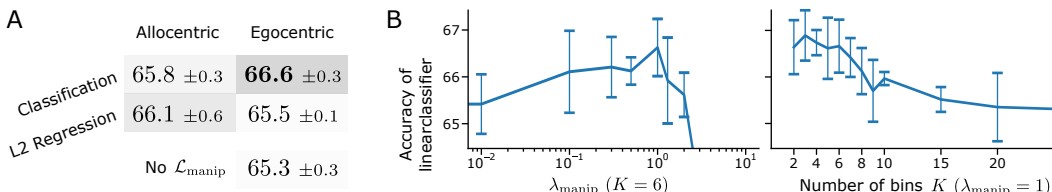

Figure 3: **Ablation study on CIFAR-100 with ResNet-18. A)** Instantiation of the manipulation-related inverse model and action representation. **B)** Hyperparameter sweep. For any setting we report mean and 95% confidence interval based on $\geq 5$ independent runs.

only loss functions are changed and manipulation-related inverse models are added. As mentioned in Section 4.1, the identity-related inverse model can also be instantiated based on other methods for SSL, such as ReLIC (Mitrovic et al., 2021) through suitable choice of $q_\theta(a_{\mathrm{id}}|x, x')$, see Appendix D. We denoted this specific variation with S-TEC* (i.e. target networks etc.).

**CIFAR-10/100 and STL-10.** We report the accuracies that linear classifiers could attain after SSL on the respective datasets in Table 1. Using 5 independent runs for each setting that we considered revealed that the manipulation-related inverse model in the case of S-TEC or S-TEC* consistently increased the accuracy of a linear classifier over the respective baseline.

Table 2: Comparing with the results of (Lee et al., 2021) on STL-10, † see Table 7 thereof.

|  | 200 Epochs | 1,000 epochs |
| --- | --- | --- |
| SimSiam impl. by Lee et al. (2021) | 86.32† | 90.2 ±0.3 |
| SimSiam + AugSelf (Lee et al., 2021) | 86.03† | 90.8 ±0.2 |
| SimCLR repr. (Chen et al., 2020a) | 86.1 ±0.2 | **91.5** ±0.4 |
| S-TEC (ours) | 86.2 ±0.2 | **91.6** ±0.2 |

We also compared our approach with the results of Lee et al. (2021), who considered a similar augmentation-aware training setting, that was mainly focused on the transferability of representations between domains. We considered the case in which their method exhibited the strongest improvement on STL-10, see Table 6 of (Lee et al., 2021) ("crop"), and retrained their model using the same number of epochs (1,000). We used their implementation and employed the same image augmentations as we did. Results are shown in Table 2. Conversely, we also tested our methods in a 200 epoch training budget, as originally done by Lee et al. (2021) and included the best reported performance that they obtained, see Table 7 of (Lee et al., 2021), noting that our method did not outperform in this case.

To investigate the differences between our approach and that of (Lee et al., 2021), we conducted an ablation study exchanging the egocentric action representation that we used with an allocentric one. In addition, we probed the impact of replacing action classification with L2 regression. Experiments were performed on CIFAR-100 using ResNet-18s, with Egocentric+Classification yielding **66.6**% accuracy over the next best setting Allocentric+L2 Regression with 66.1%, which was employed by Lee et al. (2021) (Fig. 3A and Appendix E). We hypothesize that classification affords the model more flexibility in its output distribution, thus it can handle uncertainty of its action prediction better.

**ImageNet.** We experimented with ImageNet ILSVRC-2012 (Russakovsky et al., 2015) to demonstrate that S-TEC and S-TEC* also scale. We performed training for either 100 or 300 epochs and report the results in Table 3, confirming that S-TEC is not restricted to small datasets.

Table 3: ImageNet results (ResNet-50).

| Method (100 epoch) | Top-1 (val.) |
| --- | --- |
| SimCLR repr. (Chen et al., 2020a) | 64.6 |
| ReLIC repr. (Mitrovic et al., 2021) | 66.2 |
| S-TEC (ours) | 64.8 |
| S-TEC* (ours) | **66.3** |

| Method (300 epoch) | Top-1 (val.) |
| --- | --- |
| ReLIC repr. (Mitrovic et al., 2021) | 70.0 |
| S-TEC* (ours) | **70.2** |

| Method ($\geq$ 800 epoch) | Top-1 (test) |
| --- | --- |
| MoCo v2 (Chen et al., 2020c) | 71.1 |
| SwAV (Caron et al., 2020) | 75.3 |
| SimCLR (Chen et al., 2020a) | 69.3 |
| BYOL (Grill et al., 2020) | 74.3 |
| ReLIC (Mitrovic et al., 2021) | 74.8 |
| ReLICv2 (Tomasev et al., 2022) | **77.1** |
| VICReg (Bardes et al., 2022) | 73.2 |

Table 4: Transfer learning on PASCAL VOC.

| Method (300 epoch) - Obj. Detection (AP50) | |
| --- | --- |
| ReLIC repr. (Mitrovic et al., 2021) | 82.3 (test2007) |
| S-TEC* (ours) | **82.5** (test2007) |

| Method (300 epoch) - Segmentation (mIoU) | |
| --- | --- |
| ReLIC repr. (Mitrovic et al., 2021) | 69.9 (val2012) |
| S-TEC* (ours) | **70.5** (val2012) |

| Method (1000 epoch) - Segmentation (mIoU) | |
| --- | --- |
| BYOL (Grill et al., 2020) | 76.3 (val2012) |
| ReLICv2 (Tomasev et al., 2022) | **77.9** (val2012) |

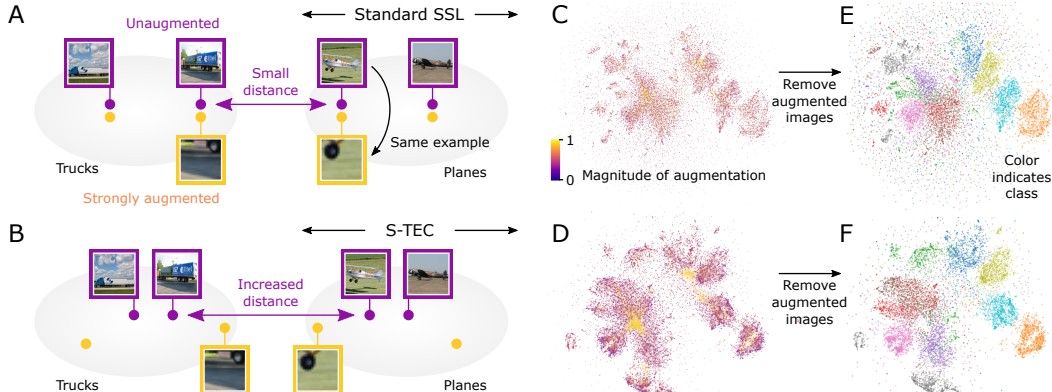

Figure 4: **Analysis. A-B**) Schematic of conjectured feature space organization: EC-unaware SSL co-locates views of an object (A), while S-TEC separates according to the level of augmentation (B). **C-D**) UMAP embeddings of images and their augmentations. Collapsed representations become separated. **E-F**) UMAP embeddings of unaugmented images, using the projection of C-D. Increased class-separation is visible.

**Object detection and semantic segmentation.** SSL aims to install generally useful representations. Thus, we considered downstream tasks beyond classification: object detection and semantic segmentation on PASCAL VOC (Everingham et al., 2010) using Faster R-CNN (Ren et al., 2015) and fully convolutional networks (Long et al., 2015), respectively, along with a ResNet-50 backbone. We initialized this backbone with the parameters that resulted from SSL on ImageNet for 300 epochs, and then fine-tuned the network on the new task (4 runs with different initializations of remaining parameters). Results are reported in Table 4, see also Appendix B and C for details.

**Hyperparameter dependence and learning dynamics.** To assess the dependence of our results on hyperparameters, we carried out several studies on CIFAR-100 with ResNet-18s: We performed a sweep over $\lambda_{\mathrm{manip}}$ that scales the impact of the manipulation-related loss, and a sweep over the number of bins $K$ used in the classification for action components. Performance depends significantly on $\lambda_{\mathrm{manip}}$, but is less affected by $K$, as long as there are not too many bins (i.e. $K < 10$), see Fig. 3B for results. Lastly, we also exhibit the loss dynamics and learning progress in Appendix E.3.

**Audio (LibriSpeech).** Finally, we also attempted to improve representation learning in the audio domain. Specifically, we considered the same data and model as introduced by Oord et al. (2018). In addition to the time-sensitive identity-related inverse model, as it emerges with CPC, we also added a time-insensitive identity-related inverse model. This allowed us to achieve **65.4**% accuracy on phoneme classification (with frozen features) as opposed to 65.1% that we obtained with CPC.

## 5   Analysis and intuitions

**Increased information content.** A consequence of learning a manipulation-related inverse model is that additional information must be expressed by the feature extractor $f$. Curiously, it had been shown in several other works that strong methods for SSL only perform well on downstream tasks, such as classification, if *intermediate* representations are used. E.g. Chen et al. (2020a) showed that inserting a nonlinear MLP between the feature extractor $f$ and the loss for contrastive SSL resulted in significantly better performance on subsequent linear classification, as opposed to the control case where this MLP was missing. This effect is explained by loss of information that is not necessarily important for the contrastive SSL objective, but for downstream tasks. In fact, recent work (Chen et al., 2020b; Mitrovic et al., 2021) observed better performance if the depth of MLP was further increased. This supports the viewpoint that additional information, albeit being potentially redundant to the contrastive SSL objective, is desired for downstream tasks of interest, see also Lee et al. (2021).

**Better organization of class borders.** Furthermore, we conjecture that learning the additional manipulation-related inverse model using $\mathcal{L}_{\mathrm{manip}}$ for S-TEC (see Eq. (2)) encourages the feature

space to be better organized. Firstly, note that conventional (contrastive or non-contrastive) SSL promotes representations of the same object in different views to be co-located (Wang and Liu, 2021) (see Fig. 4A, e.g. truck and its tire). On the other hand, S-TEC, due to its EC-aware learning, encourages representations of different views of one object (e.g. full truck vs tire) to take different positions (truck and tire in Fig. 4B). As a result, we hypothesize, the representations of canonical, untransformed views of the same type of object (e.g. trucks) must become more concentrated, to allow the transformed ones to spread. This must then increase the separation between clusters of untransformed objects (Fig. 4: purple arrow, A vs B). Moreover, S-TEC's separation of augmented views from unaugmented ones within an object-class allows the model to instead locate similar augmented views of different object classes. This then forms arguably semantically meaningful class-borders and transitions (Fig. 4B, truck tire and plane tire).

Experimentally, this hypothesis is supported by the features computed by a ResNet-50 on images of the testing set of CIFAR-10, including also their augmentations. In Fig. 4C-F, we computed lower-dimensional projections by the means of UMAP (McInnes et al., 2018), and colour-coded the magnitude of augmentation (defined as 1 minus the relative area of the cropped image). Comparing Fig. 4C and D, the model trained with S-TEC (Panel D) is aware of the zoom level of the augmentation and places more augmented images in similar regions, i.e. the borders, while the model trained without the full EC (SimCLR in this case) is oblivious to it (Panel C). If we embed the original unaugmented images in the same projection (Fig. 4E-F), the apparent class centre distance increases for S-TEC (Panel F), due to the now missing augmented images on the border. Quantitatively, we computed for each class separately the distance between the centroid of augmented image representations and the centroid of unaugmented image representations. Averaged over all classes, we find that this distance is 10.5 for S-TEC and 0.4 for SimCLR (both latent spaces cover similar scales). This further confirms that SimCLR clusters these subsets (augmented and unaugmented images) of one class around a single centroid, whereas S-TEC separates them.

Importantly, S-TEC may offer a new method for avoiding **representational collapse** (Grill et al., 2020; Bardes et al., 2022; Balestriero and LeCun, 2022) in non-contrastive SSL, because it explicitly displaces representations of the same object if they correspond to different manipulations. We have shown how S-TEC's theoretical framework recovers non-contrastive learning and extends it with a manipulation-related inverse model, however experiments are left for future work.

# 6   Related work

**SSL through auxiliary tasks.**   The idea of SSL by solving high-level queries about input manipulations was considered previously. E.g. Doersch et al. (2015); Noroozi and Favaro (2016) proposed to transform input images into patches and attempted to solve context prediction and jigsaw-puzzles respectively, while others found it useful to predict a prior rotation transformation (Gidaris et al., 2018). In contrast to such spatial prediction tasks, a different line of work by Zhang et al. (2016) discovered that colourization of black-and-white images also creates useful features for downstream tasks. Since aforementioned auxiliary tasks are orthogonal at large, prior works studied combinations and/or extensions of those (Doersch and Zisserman, 2017; Zhang et al., 2019).

**Contrastive SSL.**   Opposing to the preceding strategies of training on (handcrafted) auxiliary tasks are algorithms that originated from the idea of mutual information (MI) maximization between the input and representations thereof. In particular, the prevalent strategy of mini-batch training rendered it practical to compare – and contrast – representations of different and related inputs, which enables the maximization of MI (Gutmann and Hyvärinen, 2010). Following this perspective, algorithms for deep networks were introduced (Oord et al., 2018), and further progress ensued, where the common blueprint for the algorithm is to transform input into pairs and to contrast those against other, unrelated ones (Henaff, 2020; He et al., 2020; Chen et al., 2020a,b, 2021). Tschannen et al. (2020) opened a discussion whether it works well due to maximization of MI, which resulted in the search for different explanations, e.g. through causal interventions (Mitrovic et al., 2021; Tomasev et al., 2022). On the other hand, the profound utility of contrastive SSL has also inspired other research that attempts to connect it to hypothesized learning mechanisms in the brain. For instance, Illing et al. (2021) show that contrastive SSL can give rise to deep representations with local learning rules.

**Non-contrastive SSL.** Several other studies explored non-contrastive avenues for SSL, which have gained more traction due to their attractive properties, such as not needing negative examples. In general, these approaches require the network to produce consistent representations under content-preserving input transformations while addressing the problem of representational collapse (Grill et al., 2020; Chen and He, 2021; Zbontar et al., 2021; Bardes et al., 2022). Furthermore, Caron et al. (2020) demonstrate that enforcing consistency between cluster assignments can install potent feature extraction capabilities into a model, while also not requiring pair-wise contrasting.

**Augmentation-aware self-supervision.** Algorithms based on contrastive SSL typically aim for representations that are invariant to input transformations. While this seems appropriate in principle, subsequent studies have shown that this is not always favourable, as this strategy can exclude certain information from representations that could otherwise make them more useful for varying downstream tasks (Xiao et al., 2021). Based on similar arguments, Lee et al. (2021) proposed to predict differences of certain transformation parameters in addition to training on a standard SSL objective (Chen et al., 2020c,a; Chen and He, 2021) to improve the transferability of learned representations to other domains. These are promising results; however, on the main performance tests of SSL, i.e. testing the representations in the same domain as the training domain, these prior augmentation-aware approaches have not achieved the same performance advantage as compared to S-TEC, see Table 2.

**The relation of our approach to prior work.** Our approach generalizes methods that pair representations of paired inputs into a framework that also introduces semantic structure between paired inputs, based on the known transformations between them. This generalized and unified framework emerges from the concept of ECs and its relation to inverse models. Therefore, our approach is augmentation-aware, but its foundation on sensory-motor principles and neuroscience instructs important elements (Section 3) that are missing from earlier augmentation-aware approaches (Xiao et al., 2021; Lee et al., 2021), but have been discussed analogously by Mineault et al. (2021), and studied in part by research on local learning (Illing et al., 2021). Currently, contrastive SSL is one of the dominant approaches in the literature, and our approach improves it (and can be combined with further improvements, e.g. ReLIC) in our tests (Tables 1 and 3). Our theoretical framework also recovers and extends non-contrastive approaches, such as BYOL (Section 4.1).

## 7 Conclusion

S-TEC is a theoretical framework derived formally from first principles of biological sensory-motor control. It unifies and extends theoretically several SSL approaches, and improves them practically. Interestingly, designing S-TEC's details in a biologically-principled way is crucial for performance. S-TEC is consistently better over several strong baselines in image classification, segmentation, and object detection. S-TEC as a framework provides a new angle for future further improvements to SSL. By following established biological principles, S-TEC feeds back to neuroscience. Our results suggest that the availability of ECs to the nervous system for inverse-model learning may positively impact sensory skill. This hypothesis is testable. It predicts that subjects, exposed to a motor learning task in a novel sensory environment through active movements, would perceive the new environment better than participants that only experience passive exploration of the environment. Supporting evidence from kittens and humans already exists (Held and Hein, 1963; Bach-y Rita, 1972). S-TEC's biological implications could be strengthened even within the computational setting, by using optimization algorithms that are more biologically plausible than backpropagation. Such options have recently been described, including within SSL (Illing et al., 2021). Adding further detail to S-TEC's neural networks, such as spiking neurons, could further enhance its biological relevance.
**Limitations.** Even though our theoretical framework includes and extends various SSL methods, such as the very recent ReLICv2 (Tomasev et al., 2022), as well as non-contrastive SSL, e.g. BYOL (Grill et al., 2020), experimentally we have only extended the methods of Chen et al. (2020a); Mitrovic et al. (2021); Oord et al. (2018). In addition, we have employed only the basic commonly used augmentations for SSL without exploring other actions/augmentations. Moreover, we have experimented only with ResNets (He et al., 2016). Future research could test the advantage of S-TEC in other architectures, e.g. with self-attention (Vaswani et al., 2017).
**Potential negative societal impacts.** SSL can exploit unlabelled data, and S-TEC's rich feedback from ECs improves it, thus significantly expanding also the malicious applicability of ML. One concern is of privacy, i.e. S-TEC might assist the profiling of individuals from anonymized data.

## Acknowledgments and Disclosure of Funding

This work is partially supported by the Science and Technology Innovation 2030-Major Project (Brain Science and Brain-Like Intelligence Technology) under Grant 2022ZD0208700. The authors would like to thank Lukas Cavigelli, Renzo Andri, Édouard Carré, and the rest of Huawei's Von Neumann Lab, for offering compute resources.

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
