# OpenReview forum: "Self-Supervised Learning Through Efference Copies"
_NeurIPS.cc/2022/Conference — NeurIPS 2022 Accept_

### Official Review · Reviewer_3AEo · 2022-07-10

**Rating:** 7
**Confidence:** 4
**Soundness:** 3 good
**Presentation:** 4 excellent
**Contribution:** 3 good

**Summary:**

The authors proposed to generalize the recent self-supervised learning (SSL) methods with the Efference Copy (EC) concept in neuroscience. The proposed principles of EC lead to another implementation/variant of the SSL method which is named S-TEC, which has a more generalizable loss function and richer feedback. The new model S-TEC is benchmarked with other SSL methods, where the authors showed that it leads to improved performance overall. The authors also conducted additional latent space analysis that shows S-TEC tends to provide more separated feature space.

**Questions:**

1. As currently there are more qualitative visualizations of the latent space, it would be really interesting if more quantitative analysis could be conducted to investigate how the latent space structure is affected (e.g. how strongly augmented images distribute compared to the others).

2. The mathematical notations in the Formalism section could be more self-explainable without referring to the appendix. For example, line 181 'φ compares the outputs of f', it is unclear which type of function is used and how the outputs are compared.

3. The related work section could be put after the Analysis section for better readability.

4. The authors could include the performance trajectory of S-TEC along the training epochs to show if the model learns faster when compared to other models.

5. The authors could also include ablation experiments based on different augmentations to show if the model learns better when fewer/weaker augmentations are used.

**Limitations:**

The authors addressed some limitations of the work. Based on the suggestions and questions above, more limitations could be included after the rebuttal process.

**Strengths And Weaknesses:**

**Strength**:

1. The proposed methodology/assumption is interesting and novel.
- The authors made the theoretical connection between the EC and the SSL frameworks, which is particularly a suitable insight for the Neurips audience.
- The resulting formulation of the loss function is novel and general, which indeed provides new insights into the SSL domain.

2. The experiments seem to be nicely carried out.
- Although there are weaknesses in the experiment sections, I would like to point out that the experiments overall are nicely conducted. The method is tested on the major datasets that are used to evaluate SSL methods, ResNet-18 and ResNet-50 are both used, and the results are compared with a few major benchmark models. More could be done, but the overall experiments seem relatively convincing (under the assumption that the authors do not have an extensive amount of compute to use).
- The latent space analysis is interesting and insightful.

3. The paper is clear and nicely written.

**Weakness**:

1. Methodology: while the neuroscience connections give additional insights into the SSL framework, the actual implementation of the loss function ends up having many similarities/connections with previous works. Although the ‘The relation of our approach to prior work’ partly addressed the issue, it would be great if the authors could specifically remark how their method could or could not be reduced to other SSL frameworks.

2. Experiments could be more extensive:
- More models could be benchmarked, especially widely used ones such as MoCo and BYOL.
- ImageNet results seem to be lower than the published performance of the SimCLR (69.3% as in their paper, with ResNet-50). The details of this should be more carefully addressed.
- The 200 epoch results of SimSiam are used to compare with the 1,000 epoch results of SimCLR and S-TEC, which is biased and should be replaced with a fair comparison.

3. Others:
- All figure captions (especially Figure 1 and 2) could be significantly improved. They are not self-explainable enough without carefully reading the main text.
- The definition of the regularization loss was put into the Appendix, while it might be important to put some definition in the main text to help the readers better understand how it would affect the performance. Also, there should be some ablation experiments to evaluate how the weights of the regularization would affect the model performance.
- Line 341-342 appeared twice (also in line 293-294): ‘Moreover, we also confirmed that the benefits of our approach transfer to other, orthogonal improvements, such as provided by ReLIC, see Table 3.’

If the above weaknesses could be addressed, especially the weaknesses regarding the experiments, I am willing to raise the score.

---

> ### Author Response · Authors · 2022-08-02
> **Response to Reviewer 3AEo**
>
> We would like to thank the Reviewer for recognizing the strengths of our work, and especially for the comments and suggestions for improvement. Below we list the Reviewer’s key mentions of weaknesses, questions, and limitations, paraphrased for conciseness.
>
> > Differences from other SSL frameworks
>
> The common framework of SSL is to optimize representations such that these become oblivious about the augmentations that were applied, i.e. the common strategy is to produce invariant representations across augmentations.
> Example methods that follow that direction include SimCLR, MoCo, BYOL, ReLIC, etc.
> This is in contrast to our methodology, as we instead pose the learning problem as a general question: what action was applied?
> Because of this, our method cannot be reduced to the aforementioned framework(s), but rather, we can reduce one part of our approach, i.e. specifically the identity-related inverse model to the previously mentioned methods.
> Our approach, i.e. S-TEC, is a broader framework in which these approaches can be framed, and a framework that extends each of these approaches.
> We have shown this with detailed derivations in the Appendix D.
> We recognize that this aspect of rigor in our work may not have been emphasized enough in the previous version of the manuscript.
> In the revised manuscript we have added more explicit pointers to the Appendix, and have reorganized the results section to also include a summary of those theoretical results.
>
> In summary, S-TEC's design choices are derived quite rigorously and from first principles of the biological nervous system and of sensory-motor control.
> This point of origin leads to S-TEC as a framework that is rather general (recovers various existing SSL methods, extends them, and can potentially generate new ones), biologically-insightful, modular (S-TEC's realization of the manipulation-related inverse model can be kept fixed while its identity-related inverse model can vary), and formal (we have provided detailed derivations of the relation to various SSL methods).
>
> > More models could be benchmarked, especially widely used ones such as MoCo and BYOL.
>
> We agree with the reviewer that more benchmark comparisons would be insightful, hence we have conducted additional experiments with the proposed methods of MoCo and BYOL in the ResNet-18 setting.
> For these methods, we optimized the target network update factor as well as the dimensions of the hidden layer of $g$ (identity-related inverse model), see Table 1 in the revised manuscript.
>
> > Fair comparison (200 epochs, 1,000 epochs)
>
> In terms of comparison to the work of Lee et al. (2021), we have conducted additional experiments to provide a fair comparison: We obtained the published code of Lee et al. (2021) and retrained their models for 1,000 epochs.
> Likewise, we have retrained our models for 200 epochs and report both settings in Table 2 in the revised manuscript.
> We find that S-TEC outperforms AugSelf in the 1000-epoch case, but did not do so in the 200 epoch budget, which we point out in the revised manuscript.
>
> > Low SimCLR ImageNet performance
>
> In regard to the performance of SimCLR on ImageNet, we would like to comment that the result of 69.3% by Chen et al. (2020) is reportedly obtained with 1,000 epochs of training, while 100 epochs of training yield ~64.6%, as judged by Figure 7 of the respective publication (Chen et al., 2020), which would indeed match our reproduced result.
>
> > Others
>
> We appreciate the comments on the writing and have attempted to address these items.
> Specifically, we have updated our Figure captions, removed duplicated sentences, attributed mathematical notations with further comments and swapped the related work section with the analysis section.
> Furthermore, we include training loss curves in the Appendix E.4 (Figures S4-S9) and point towards them in the main text.
>
> > more quantitative analysis on latent space structure
>
> In response to this item, we considered how strongly augmented images distribute compared to unaugmented ones, following the Reviewer's suggestion.
> Specifically, we computed for each class separately the distance between the centroid of augmented image representations and the centroid of unaugmented image representations.
> We find that this distance, averaged over classes, is significantly larger for S-TEC, i.e. 10.5 as compared to what we obtained with SimCLR: 0.4, despite similar latent space scales (overall variance of the features).
> This quantitative analysis confirms that SimCLR clusters augmented and unaugmented images of one class around a single centroid, whereas S-TEC separates them.
> We included this result in a new paragraph in the analysis section of the revised manuscript.
>
> All in all, we believe that we have addressed the Reviewer’s key concerns and we look forward to the updated feedback.

---

> > ### Comment · Reviewer_3AEo · 2022-08-05
> > **Improve score**
> >
> > My major concern about the paper is addressed nicely. The authors made sufficient and important revisions to the paper that improved the paper quality by quite a lot. I (personally) might suggest the authors write a general response that highlights what is been improved.
> >
> > I am aware that other reviewers raised concerns about the paper's novelty, however, I stand by my judgment about how the proposed framework is relatively novel mainly because of how it is derived from neuroscience principles, which is not only conceptually insightful, but also theoretically insightful from the neuroscience perspective.

---

> > > ### Author Response · Authors · 2022-08-08
> > > **Response to Reviewer 3AEo**
> > >
> > > We would like to thank the Reviewer 3AEo again for the valuable comments that helped us improve the manuscript, and for acknowledging the significance of our work.

---

### Official Review · Reviewer_wiNh · 2022-07-10

**Rating:** 5
**Confidence:** 3
**Soundness:** 2 fair
**Presentation:** 3 good
**Contribution:** 2 fair

**Summary:**

Inspired by the concept of Efference Copy (EC), the authors propose a new self-supervised learning (SSL) framework, called Self-supervision Through Efference Copies (S-TEC). Specifically, the authors focus on inverse models of EC, which map sensory inputs to motor outputs, and make the analogy between EC and SSL by considering data augmentation as actions. The inverse models are implemented by the identity-related inverse model to determine whether two inputs are the same object or not, and the manipulation inverse model to figure out which augmentation is applied. Through the experiments, the authors demonstrate that S-TEC outperforms the reference algorithms including SimCLR and AugSelf on the image classification tasks, and analyze why S-TEC works well in a qualitative manner.

**Questions:**

- Please give more details about the egocentric representation of actions in lines 140-151.
- Please clarify the difference with [1] in the experiments. For now, the comparison is done on only STL-10 and the number of training epochs for S-TEC is larger than [1], which makes me suspicious about whether the performance improvement comes from S-TEC or not. If the main difference between S-TEC and [1] is the egocentric representation of actions, please conduct an ablation study of S-TEC by replacing the egocentric representation of actions with the allocentric one as [1].
- Please conduct ablation studies about the manipulated-related inversed model, such as varying the number of clusters.
- Please conduct additional experiments on various tasks other than image classification.

**Limitations:**

Yes.

**Strengths And Weaknesses:**

Originality: The motivation of this paper that brings the concept of EC into SSL sounds interesting. However, the novelty is a bit limited because the authors do not consider forward models that are critical parts of EC in S-TEC and it is difficult to see the significant differences from previous studies. The loss function of the identity-related inverse model is almost identical to a conventional contrastive loss used in SSL literature and the loss function of the manipulated-related inversed model is similar to [1].

Quality: The authors' claims are not supported well. First, the comparison with [1] is not enough. Since both the proposed method and [1] are motivated by augmentation-aware SSL, the two methods should be compared intensively. However, the authors compare the two methods on only STL-10 and the experiment setups are not identical. Second, the ablation study is lacking completely. For example, it would be great to see the effect of the number of clusters for the manipulation-related inverse model. Finally, the experiments are done on only image classification tasks. Considering the purpose of SSL is to learn general features for various downstream tasks, the authors should conduct more experiments on various tasks, such as object detection.

Clarity: The paper is generally well-written and easy to follow. However, some important details are in the supplemental material not in the main text.

Significance: At this stage, the significance of this paper is limited because it is not clear what are the contributions compared with previous work and the experimental results are not enough to support the authors' claims as I wrote above.

Reference\
[1] Lee, H., Lee, K., Lee, K., Lee, H., and Shin, J. (2021). Improving transferability of representations via augmentation-aware self-supervision. Advances in Neural Information Processing Systems, 34.

---

> ### Author Response · Authors · 2022-08-02
> **Response to Reviewer wiNh**
>
> We would like to thank the Reviewer for recognizing the strengths of our work, and especially for the comments and suggestions for improvement.
> Below we list the Reviewer's key mentions of weaknesses, questions, and limitations, paraphrased for conciseness, and we address each point.
>
> > limited novelty compared to AugSelf, Comparison with AugSelf not enough, ablation study is lacking completely
>
> - In response to the Reviewer's concerns relating to the connection to the work of Lee et al. (2021), we would like to point out that our manipulation-related loss uses a different action representation, and on discrete bins rather than continuous values.
> We show systematically in the revised manuscript that these differences are important for performance.
> Moreover, an important difference is that these design choices are not arbitrary but rather are grounded on biological principles, that turn out to also be beneficial.
> Specifically, Lee et al. (2021) consider to predict differences of augmentation parameters such as the difference in the center point of the image crop, and the difference between the crop's distances to the original image border.
> These differences have the original image as a reference frame, which is why we refer to them as "allocentric".
> In contrast, we compute a transformation matrix that connects the views in a relative manner, and which can be readily viewed as an actual action represented in the reference frame of the perceiving instance, i.e. the person taking the action, or the ResNet in our case.
> Hence, we denote this as an "egocentric" representation.
> This choice was dictated by our framework's point of origin in embodied sensory-motor control, where an agent does not merely observe augmentations but actively takes own actions.
> In addition, we chose to instantiate the manipulation-related inverse model as a classifier, which gives it more flexibility in its output distribution of the action, and hence allows the model to deal better with uncertainty in that prediction.
> This choice is also directly dictated by biological constraints, according to empirical evidence that suggests the brain represents actions as discrete clusters.
> - To verify the importance of these design choices, we have added to the revised manuscript corresponding ablations studies that remove the classification, as well as the egocentric representation in Appendix E.2 (Table S3), and refer to them in the main text as well.
> In addition, we have added a sweep over the number of clusters, finding that the performance is high as long as the number of clusters does not become too large, see Figure S3 in Appendix E.3.
> Furthermore, we point out that we had also conducted a parameter sweep over the impact of the hyperparamter that weighs the impact of the manipulation-related loss, see Figure S2.
> Both studies confirm the positive impact of our choices.
> - We do, however, acknowledge that S-TEC has certain analogies to the work of Lee et al. (2021).
> Both can be viewed as "augmentation-aware" training methods, but with a different focus and realization.
> It should be mentioned that the work of Lee et al. (2021), did not focus on improving performance in the same domain as the training domain which our work does, but rather they focused on transferability of features.
> In light of this, even though our work outperforms AugSelf in our experiments, we suggest that S-TEC and AugSelf both demonstrate important new aspects of augmentation-aware learning.
>
> > the two methods should be compared intensively
>
> To compare the two methods more intensively, we used AugSelf's [1] published code, and we performed optimization using their hyperparameters for 1,000 epochs.
> In addition, we have trained with our approach for 200 epochs.
> These results are reported in Table 2 of our revised manuscript.
> S-TEC outperforms AugSelf in the 1000-epoch case, but did not do so in the 200 epoch budget, which we point out in the revised manuscript.
> See also above comment on the ablation study of the different design choices (Table S3 in Appendix).
>
> > Other tasks than classification
>
> We agree with the Reviewer that one goal of SSL is to learn general features for various downstream tasks.
> Therefore, we have conducted a study on semantic segmentation (which has also been considered in relevant work for SSL), and have added those results in Table 4 of the revised manuscript.
> S-TEC provides improvements in this case too, compared to ReLIC.
>
> We believe that we have addressed the Reviewer's concerns and we look forward to the updated feedback.

---

> > ### Author Response · Authors · 2022-08-08
> > **Response to Reviewer wiNh**
> >
> > We would like to thank the Reviewer wiNh again for the valuable comments that helped us improve the manuscript. We have taken the Reviewer's concerns and suggestions seriously, we have responded in detail in our comments, and have rigorously updated the manuscript. We believe we have addressed the key concerns of the Reviewer. As the discussion period approaches its end, we would greatly appreciate an updated feedback.

---

> > ### Comment · Reviewer_wiNh · 2022-08-10
> > **Response**
> >
> > Thanks for taking care of my comments.\
> > Although the authors addressed many of my concerns, I have a few follow-up questions:
> > > It should be mentioned that the work of Lee et al. (2021), did not focus on improving performance in the same domain as the training domain which our work does, but rather they focused on transferability of features.
> >
> > It is not clear what "domain" means. Lee et al. (2021) did various tasks in the vision domain. Furthermore, the goal of SSL including both Lee et al. (2021) and this manuscript is to learn the features that are useful for the downstream tasks from unlabelled data.
> >
> > >To compare the two methods more intensively, we used AugSelf's [1] published code, and we performed optimization using their hyperparameters for 1,000 epochs.
> >
> > Could you explain more details in Table 2? It seems that the authors reported the results of SimSiam + AugSelf written in Table 7 of Lee et al. (2021) for 200 epochs, which uses "crop", "jitter", and "solarization"; but reported the results of SimSiam + AugSelf using only "crop" for 1000 epochs.

---

> > > ### Author Response · Authors · 2022-08-10
> > > **Counter-response**
> > >
> > > > It is not clear what "domain" means. Lee et al. (2021) did various tasks in the vision domain. Furthermore, the goal of SSL including both Lee et al. (2021) and this manuscript is to learn the features that are useful for the downstream tasks from unlabelled data.
> > >
> > > The paper of Lee et al. was entitled “Improving *Transferability* of Representations via Augmentation-Aware Self-Supervision” (emphasis our own). That work was mainly concerned with Transfer Learning, where learning occurs in one domain and the learned features are tested in another. Specifically, in that paper's case, the different domains were different datasets. On the other hand, the paper did not focus on improving performance on the same domain (i.e. dataset), even though that did turn out to be the case. For example SimSiam + AugSelf surpasses SimSiam, when all training and testing is on STL-10. In our work, we focused on testing within the same domain as training, and we showed that S-TEC improves performance compared to other SSL techniques, including AugSelf.
> > > We did not focus on S-TEC's possible transferability improvements, even though that is certainly an interesting focus of a possible future study. We did provide evidence for improved transfer learning, by applying S-TEC's learned representations not only to image classification, but also to semantic segmentation of images. A model trained with S-TEC performed better than ReLIC in both tasks. Nevertheless, this is the only related experiment we performed, as transferability was not the focus of our work here.
> > >
> > > We hope that this clarifies the difference in focus between our work and Lee's et al.'s.
> > >
> > >
> > > > Could you explain more details in Table 2? It seems that the authors reported the results of SimSiam + AugSelf written in Table 7 of Lee et al. (2021) for 200 epochs, which uses "crop", "jitter", and "solarization"; but reported the results of SimSiam + AugSelf using only "crop" for 1000 epochs.
> > >
> > > In brief, using all three augmentations ("crop", "jitter", and "solarization") but being aware of only “crop” is the best setting for SimSiam + AugSelf, hence this is what we report in our Table 2 to be fair to Lee et al.’s (2021) work.
> > >
> > > To elaborate, the best 200-epoch result (86.03%) reported in Lee et al. (2021) was from a model that was trained using all three augmentations ("crop", "jitter", and "solarization") (see Table 7 of Lee et al. (2021)), but was aware of only "crop" and "jitter".
> > >
> > > Predicting all three augmentations deteriorated SimSiam+AugSelf's performance (85.91%). This can be also seen in Table 7 of that work.
> > > Table 7 did not report a result of a model aware only of “crop”, but it can be seen elsewhere in the Lee et al. (2021) paper and in our own control experiments that this is the best setting.
> > >
> > > Specifically, first, Table 6 of Lee et al., which uses weaker augmentations than their Table 7, i.e. “crop” and “jitter”, without “solarization”, shows that predicting only “crop” yields the best result (85.98%) in this setting of weaker data-augmentation, better than predicting both "crop" and "jitter" parameters (85.70%). However, the strongest results in the paper were achieved by also applying (but not predicting) “solarization” (Table 7). Taking these together, Lee et al.’s results suggest that using all augmentations but only predicting “crop” would be the best setting, but they did not report this.
> > >
> > > Second, in our 1000-epoch experiments with SimSiam+AugSelf we used the augmentations that yielded the best results (with 200 epochs) in Lee et al. (2021), i.e. including “solarization”, and we experimented both with their best-reported setting (predicting “crop” and “jitter”) and their best-indicated but not reported setting (predicting only “crop”). We achieved 90.8% accuracy with the “crop”-only-aware setting, higher than 88.7% that was reached when the model was also aware of the “jitter” augmentation.
> > >
> > > Hence, for a fair comparison, in our Table 2 we report the “crop”-only-aware result of AugSelf, because this is the best setting both according to our experiments, and according to what can be inferred from Lee et al.’s reported results. To be sure, the experiments used data with all three augmentations, including “solarization”.
> > >
> > > We will add a clarification on this to the camera-ready manuscript.
> > >
> > > > the authors addressed many of my concerns
> > >
> > > We would like to thank the Reviewer for acknowledging this, and for providing a response. The Reviewer's constructive suggestions helped us substantially improve the manuscript.
> > >
> > > We believe that our previous rebuttal and our response here address all concerns of the Reviewer, and we hope that this can be reflected in an update of the Reviewer's evaluation of our work.

---

### Official Review · Reviewer_bN43 · 2022-07-11

**Rating:** 5
**Confidence:** 3
**Soundness:** 2 fair
**Presentation:** 2 fair
**Contribution:** 2 fair

**Summary:**

The paper states a broad framework for learning in a setting where actions lead to changes to the environment/input, via the (neuroscience-inspired) notion of an efference copy, in which the agent gets a copy of the action ("motor command") in addition to sending it off to the motor system. Using a vision setting, and by setting up two simple actions, "look at the same object or not" and "change the view (affine transformation) of the object", the paper derives a loss function which is identical to a typical contrastive learning objective, plus a term that handles the case of change of view, for which they use data augmentation. The paper shows that this framework yields representations that are better for linearly classifying in several visual tasks.

**Questions:**

Can you explain concisely why the notion of efference copies is particularly valuable for this example or for other possible examples?

**Limitations:**

The framework presented is so general that it is a bit tricky to think of ethical implications, but given that the paper rotates around one specific example, I would have suggested the authors dig a little more into the object or visual classification example to see what negative impacts they can think of of either better features, or perhaps the specific model described here (e.g. would it be possible to use this model to get better predictions of held-out views of objects or faces?)

**Strengths And Weaknesses:**

Strengths

This framework is very interesting, and raises lots of obvious questions about what would happen if one were to change the set of actions. In particular, I am curious how this could be applied to speech. Typical approaches are not merely contrastive, but predictive, i.e. they have a temporal dimension (wav2vec2, CPC), so there would perhaps be some way of modifying the current approach, or perhaps a more radical approach in the same framework could be developed that takes an analysis-by-synthesis angle.

The results obtained are somewhat better than the reference systems.

Weaknesses

I think the notion of efference copy is not really doing anything here, aside from re-framing the problem, given that we are assuming that the efference copy is always exactly accurate. As such, given that there is only one example given (the specific cashing-out of the framework for vision), and given that there are no ablation studies or error bars, it is not so clear to me what we are supposed to get out of the result. Is the notion of efference copy concretely useful to improve performance? Or is it merely the combination of contrastive learning with data augmentation? Or is it not either which is in fact improving the result over previous scores (perhaps implementation differences, random initialization, etc)? In other words, to clearly establish the value of this new framework, I would expect one of two things: more, diverse examples; or a detailed and careful analysis of one example. The discussion around Figure 3 would imply that the authors are suggesting that it is the combination of contrastive learning with data augmentation which is helping, but that is not really a property of the general framework from what I can see, but rather just of this specific example and the actions that the authors have chosen. Maybe this could be okay, if the paper were reframed as being less about the efference copy framework (which doesn't seem to have any specific value here) and more about combining contrastive SSL with augmentation, but by itself that seems like a small extension with not such big empirical improvements.

---

> ### Author Response · Authors · 2022-08-02
> **Response to Reviewer bN43 1/2**
>
> We would like to thank the Reviewer for recognizing the strengths of our work, and especially for the comments and suggestions for improvement. Below we list the Reviewer's key mentions of weaknesses, questions, and limitations, paraphrased for conciseness, and we address each point.
>
> > Discuss: "efference copy is not really doing anything aside from reframing"
>
> The notion of the efference copy allowed us in this case to derive "augmentation-aware" training from first principles.
> This derivation led to a natural combination of augmentation-aware training with contrastive and non-contrastive SSL, which is itself novel.
> In addition, we would like to point out that the reframing of a problem can be useful to understand a problem from a different perspective, hence we view this as a contribution on its own.
> Indeed, it allowed us to place several important existing self-supervised learning (SSL) approaches such as SimCLR, ReLIC and BYOL in a single framework of embodied cognition, and to extend each of them with an added term.
> Furthermore, our description of the framework in these terms, and with biological and embodied constraints, enables an interpretation of our results as a potential role of efference copies in the biological sensory-motor system.
> Moreover, biological and embodied first principles led us to specific design choices for the "augmentation-aware" aspect of training, which are new and different from prior "augmentation-aware" approaches.
> These differences are necessary for the improvements that we gain (see also comment on ablation study below).
> To be concrete, besides the important conceptual implications of the efference copy, we have demonstrated performance improvements compared to important baselines.
> In the revised manuscript we have significantly strengthened the evidence for these improvements with new experiments, reinforced statistics, and ablation studies.
> The new experiments include comparisons with several baseline approaches, and they confirm and generalize the strength of our approach.
>
> > No error bars. No ablation studies.
>
> - In fact, we did report error bars in the earlier submitted Appendix, where we had also conducted an ablation over the hyperparameter that scales the contribution of the additional manipulation-related loss (Figure S2 in Appendix).
> Possibly, we have not pointed out these items strongly enough in the previous main manuscript, and as such they may have not been considered.
> We would like to thank the Reviewer for bringing this to our attention.
> In summary, the statistics and ablation studies confirm that S-TEC brings significant performance improvements.
> Due to the fact that we used the same underlying code base, as well as the same parameter initialization strategies for all our experiments, and performed 5 independent optimization runs, we are confident that the improvement is not due to random variations, but because of the contributions that S-TEC provides, as we have derived.
> In our revised version, we include error bars directly in Table 1, while also linking additional ablation studies more explicitly.
> - Additionally, we have extended the ablation studies that we conduct to determine the impact of discrete and egocentric action representations, see Table S3 in Appendix.
> Moreover, we added a further hyperparameter study that examines the impact of the number of clusters in the manipulation-related action classification, see Figure S3 in Appendix.

---

> > ### Author Response · Authors · 2022-08-02
> > **Response to Reviewer bN43 2/2**
> >
> > > more, diverse examples or a detailed and careful analysis of one example
> >
> > - In the revised manuscript we have followed both suggestions of the Reviewer.
> > Specifically, first, we have considered another example task, namely that of image segmentation.
> > We show that in this case too S-TEC outperforms ReLIC, i.e. a strong and recent SSL approach.
> > Second, we have also performed a detailed and careful analysis of the classification task, with error bars, ablation studies, hyperparameter sweeps and extensions to several frameworks, theory, and links to both ML and neuroscience.
> > - Moreover, we have emphasized in the revised manuscript the broad implications of our framework.
> > Specifically, we mention more explicitly that the Appendix includes detailed and formal derivations, which show how several important SSL approaches, such as SimCLR, BYOL and ReLIC are recovered and extended as part of our S-TEC framework from the first principles of efference copies.
> > In addition, we have reorganized the Results section (section 4) to emphasize also these theoretical results, as opposed to only the experimental results.
> > - We agree with the Reviewer that it would be interesting to extend our experiments towards different modalities such as audio and speech.
> > Due to the limited time available in the rebuttal period, we would like to sketch a blueprint for doing that and how the notion of the efference copy can indeed help to think about it: We recognize that audio is directly affected by the surrounding environment, i.e. the room response, and hence, actions such as moving around will directly affect the perceived sound.
> > To simulate this, one could consider different room response filters as augmentations, which can then be predicted by the manipulation-related inverse model for instance.
> > Further active manipulations of the sound could instead be chosen to simulate different pitches, durations, and other parameters of the vocalizations, somewhat mimicking babbling of infants, or songs of young songbirds – a common model organism for the study of vocal learning.
> > We expect that in this domain too it may be shown that S-TEC enables better learning for auditory processing.
> >
> > We believe that we have addressed the Reviewer's key concerns and we look forward to the updated feedback.

---

> > > ### Author Response · Authors · 2022-08-08
> > > **Response to Reviewer bN43**
> > >
> > > We would like to thank the Reviewer bN43 again for the valuable comments that helped us improve the manuscript. We have taken the Reviewer's concerns and suggestions seriously, we have responded in detail in our comments, and have rigorously updated the manuscript. We believe we have addressed the key concerns of the Reviewer. As the discussion period approaches its end, we would greatly appreciate an updated feedback.

---

> > > ### Comment · Reviewer_bN43 · 2022-08-09
> > > **Response**
> > >
> > > Thanks for your reply,
> > >
> > > > The notion of the efference copy allowed us ... to place several important existing self-supervised learning (SSL) approaches such as SimCLR, ReLIC and BYOL in a single framework
> > >
> > > I agree that this kind of theoretical contribution is valuable, but many of these approaches were already known to be formally similar. The more important question is whether the "augmentation-aware" setting that the authors are proposing is genuinely representative of neural dynamics in the presence of efference copies, or merely loosely analogous. This is a question that is pretty clearly beyond the scope of a paper like this one.
> > >
> > > > Due to the fact that we used the same underlying code base,
> > >
> > > This point isn't clear, either in the revised manuscript or in the response to reviewers: when comparing S-TEC against SimCLR and ReLIC, what exactly is the overlap in the code? In other words, given that, as the manuscript demonstrates, the approaches are closely related, and given that the effects are small overall, an informative comparison would demand that the difference in the code base be minimal. In this vein, it is important to stress that, from looking at the current version of the PDF, the ablation experiments (as well as the hyperparameter search which responds to some questions about the importance of the parts of the loss function) are contained, or have  in supplementary materials, and not in the main paper. These experiments remain quite important to interpreting the results, and should not be relegated to supplementary materials - not only are these not considered part of the task of the reviewer by the NeurIPS guidelines, regular readers cannot be expected to examine the supplementary materials for critical information.
> > >
> > > > we have considered another example task, namely that of image segmentation
> > >
> > > This is a helpful example, and responds to some of my objections.

---

> > > > ### Author Response · Authors · 2022-08-09
> > > > **Response to Reviewer bN43**
> > > >
> > > > We would like to thank Reviewer bN43 for his response and for engaging in a discussion.
> > > >
> > > > >> a single framework
> > > >
> > > > > I agree that this kind of theoretical contribution is valuable, but many of these approaches were already known to be formally similar.
> > > >
> > > > We thank the Reviewer for recognizing the value of unifying the ML approaches in a single framework. However, we would like to note that the theoretical value of our approach is more than that. We do not merely describe the common framework for SSL, but rather the framework emerges from well-justified first principles through our derivations.
> > > >
> > > > > The more important question is whether the "augmentation-aware" setting that the authors are proposing is genuinely representative of neural dynamics in the presence of efference copies, or merely loosely analogous. This is a question that is pretty clearly beyond the scope of a paper like this one.
> > > >
> > > > The Reviewer is correct that our work does not provide concrete insight at the level of neural dynamics. However, there are multiple levels of description of the nervous system and its function, which are certainly within the scope of neuroscience. Our work offers insights at a system level of description.
> > > >
> > > > Specifically, it shows that a sensory-motor system operating with inverse models, that has access to efference copies (EC), and is equipped with a hierarchical learning algorithm, can use the EC to learn useful sensory representations while acting on its surrounding objects.
> > > >
> > > > Previously, EC was indeed associated with learning, but specifically with the learning of motor skills. Here we show through modelling that EC is useful for acquiring sensory skills as well.
> > > >
> > > > In addition, it shows that multiple inverse models learned by exploiting the full content of ECs do not interact catastrophically but rather can act synergistically to increase the quality of the representation than using only an identity-related or only a manipulation-related EC.
> > > >
> > > > The ingredients of our model are not biologically detailed in our simulations, but they are biologically plausible. ECs, learned inverse models, learning of hierarchies, even specifically through approximations to backpropagation, actions represented as clusters, are all present in the biological sensory-motor system, and are the elements that are necessary for our framework and its results. Therefore, we suggest that there is in fact some insight from our work into the possible effects of ECs in the nervous system's function.
> > > >
> > > > Moreover, our model and its results have generated a hypothesis that is experimentally testable. It predicts that subjects exposed to a motor learning task in a novel sensory environment through active movements, would perceive the new environment better than participants that only experience passive exploration of the environment. We believe that these are meaningful contributions to sensory-motor neuroscience that cannot be discounted, even if they are not at the analysis level of neural dynamics. These neuroscience-related contributions have been already emphasized in the manuscript, and we hope they are now clearer to the Reviewer.
> > > >
> > > > > What exactly is the overlap in the code
> > > >
> > > > The overlap is indeed everything except for the loss function and the added manipulation-related inverse model. We have now explicitly clarified this in the revised manuscript (see Section 4.2.).
> > > > As a side remark, we would also like to point out that source code is included in the submitted supplementary material.
> > > >
> > > > > These experiments remain quite important to interpreting the results, and should not be relegated to supplementary materials
> > > >
> > > > We thank the reviewer for emphasizing the importance of these experiments. Therefore we have moved these results into the main text of our new revision in response to this point.
> > > >
> > > > We hope that these new clarifications and improvements to the manuscript can be considered for a follow-up feedback by the Reviewer.

---

> > > > > ### Comment · Reviewer_bN43 · 2022-08-09
> > > > > **Response**
> > > > >
> > > > > Provided these additional analyses are in the main text I am happy to improve my score a little.

---

> ### Author Response · Authors · 2022-08-09
> **Response to Reviewer bN43**
>
> >  In particular, I am curious how this could be applied to speech.
>
> We thank the Reviewer for this suggestion, and we attempted to apply our concepts to speech as well.
> Specifically, we experimented with CPC [1] using the same dataset, employing published code by [2].
>
> We installed here, in addition to CPC, an inverse model that we refer to as the time-insensitive identity-related inverse model, which is tasked to classify (binary) whether an audio feature vector z (see CPC notation) is within 120 ms of proximity to the reference feature vector c.
> This is in contrast to the timing-sensitive identity-related inverse model that emerges from the CPC method, which specifically considers whether sequence **and timing** is correct. I.e. CPC treats also feature vectors as negative that are just 10 ms off.
>
> We implemented this inverse model as the same MLP as our manipulation-related inverse models in the vision tasks (i.e. 512 hidden units), albeit with input dimension that corresponds to the concatenation of a single c vector and a single z vector and output dimension of 2.
>
> We found this approach (albeit not at all tuned) to deliver improved phoneme classification accuracy, similar to the vision tasks.
> Note however that these results are single runs, but the 300 and 1,000 epochs use different seeds.
>
> | | 300 epochs | 1,000 epochs |
> |---|---|---|
> |CPC | 60.8 | 65.1 |
> |S-TEC(CPC) | 61.3 | 65.4 |
>
> Hence, we have additionally shown that the framework of S-TEC is not only restricted to vision, but its ideas can also be forwarded to domains such as speech.
>
> ## References
> [1] Oord, A. van den, Li, Y., & Vinyals, O. (2019). Representation Learning with Contrastive Predictive Coding (arXiv:1807.03748). arXiv. http://arxiv.org/abs/1807.03748
>
> [2] Löwe, S., O’Connor, P., & Veeling, B. S. (2020). Putting An End to End-to-End: Gradient-Isolated Learning of Representations. ArXiv:1905.11786 [Cs, Stat]. http://arxiv.org/abs/1905.11786

---

> > ### Comment · Reviewer_bN43 · 2022-08-09
> > **Response**
> >
> > This is an interesting follow-up experiment - thanks, it does broaden the paper.

---

### Official Review · Reviewer_fHUb · 2022-07-12

**Rating:** 5
**Confidence:** 4
**Soundness:** 3 good
**Presentation:** 3 good
**Contribution:** 4 excellent

**Summary:**

Author propose a new neuroscience-inspired self-supervised learning process through adopting processes of the brain for developing a new framework. Specifically the use the concept of efference copy to derive a new loss function in a contrastive setting that can be used to perform self-supervised learning. They demonstrated the applicability of this method on a number of different datasets on image classification. There was a modest increase in performance when compared against some of the earlier work on contrastive SSL, i.e. SimCLR, which is actually the biggest limitation of this work. Nevertheless that is an interesting piece of work that is among the 1-2 other papers that have approached this problem from a more complex cognitive perspective.

**Questions:**

A) Did you consider comparing with other methods beyond SimCLR and Relic? It would have been positive irrespective of absolute performance
B) When comparing with methods that provided results in 200 epochs (arguably very few labs can train for longer than 200/300 epochs, as usually methods that present 1000 epochs on image net for instance are large lab and companies **usually**), one should compare their own method on 200 epochs rather than comparing the 1000 vs 200.
C) More of theoretical question, if this method can barely outperform SimCLR, what further improvements adopted from neuroscience could perhaps lead to further improvements that can compare with say VicReg?

**Limitations:**

I think the results section is a clear limitation and someone reading this paper will be confused as to what the current SOTA is. Contrastive methods have seen better days and we can safely say that most work now is non-contrastive - this has not been adequately articulated in the paper and further information should have been provided on other methods such as regularisation and VicReg, SWAV, etc.

The paper would have been extremely interesting 2 years ago, but as it is a different way of thinking I do not necessarily see this as a limitation, given that advances are not linear across sub-fields. What I mean by that is that it could be that in a year from now a neuroscience-inspired methods built upon this one could outperform SOTA in SSL. I just think that all these should have been covered in the main paper, and also not use the supplementary material to add so much information that 10-15% of which would fit best in the main paper. I literally looked at the supplementary material to get some essential information whereby essential information should be covered in the main paper, and leave additional context for the supplementary material.



**Strengths And Weaknesses:**

Strengths.

A) the paper is well founded and written. Although I am not a neuroscientist, I appreciate that the formulations used are inspired by how our central nervous system sends stimuli to our motor neuron system to perform an action, whilst also sending a signal to itself. That is grounded to the fact that our central nervous systems learns primarily without external supervision.
B) There is a parallel presented around how to recover SSL methods such as contrastive learning by inverse models for efference copy using first principles.

Weaknesses.

A) Although it seems to be foundationally grounded, it mainly focuses on comparisons with “very early” (oxymoron I know) work on contrastive learning (SimCLR), which I dare say that is not that relevant anymore, and is mainly used as a baseline when comapred with other approaches. It is common to compare with SimCLR but when including BYOL, SWAV, MoCo, etc. I think the authors should have added comparisons with all these approaches, even if this methods did not outperform them. I am not taking the side at all on “SOTA” or “nothing”   -  I mean that from the point of of completeness of the experimental design and rigourous representation around how this method compares with others.
B) Figure 1 is incomplete and notations are inconsistent but that has been acknowledged in the supplementary material and the authors have committed to fix these issues in the camera-ready version.
C) Presentation could have been stronger, i.e. additional figures could have been provided on how the model trains, loss function, etc. over those 1000 epochs. There is limited information in general on the results part.
D) Conclusion is rather weak, not only because it does not reflect on the neuroscience side more than it did, but it rather repeats and points to sections to draw conclusions. It is more like a summary/overview than a conclusion

---

> ### Author Response · Authors · 2022-08-02
> **Response to Reviewer fHUb 1/2**
>
> We would like to thank the Reviewer for recognizing the strengths of our work, and especially for the comments and suggestions for improvement.
> Below we list the Reviewer’s key mentions of weaknesses, questions, and limitations, paraphrased for conciseness, and we address each point.
>
> > SimCLR is obsolete
>
> - We acknowledge that there has been substantial progress in self-supervised learning (SSL) beyond SimCLR. However, SimCLR is a suitable starting point for our study since it is a highly impactful and common baseline.
> - Moreover, other important methods such as MoCo and ReLIC are directly related to SimCLR's contrastive loss.
> - Perhaps more importantly, our comparisons between SimCLR and a SimCLR-based realization of S-TEC showed empirically that S-TEC’s extension of SSL is not only a conceptual framework, but also has practical potential for improving existing SSL methods.
> - Our results on the SimCLR-based S-TEC do not have implications only against SimCLR itself, but also for S-TEC's potential against other methods. That is because S-TEC is directly explorable as an extension of other strong SSL approaches. About this point, please also see our further below response concerning the content of the Appendix.
> - However, we do agree that it would be best to compare also directly with more recent approaches such as BYOL and MoCo, which we have now performed and reported in the revised manuscript (see also responses below).
>
> > Comparisons with more recent SSL approaches (BYOL, MoCo). Current SOTA is non-contrastive.
> - In the revised manuscript, we have followed the Reviewer's suggestion, and we have added BYOL and MoCo baselines for ResNet-18 on CIFAR-10/100 and STL-10. For these results we tuned the target network's update factor as well as the dimension of the hidden layer of $g$ (identity-related inverse model) and we report average and standard deviation for 5 runs. In summary, S-TEC has outperformed those approaches too (see Table 1 of the revised manuscript).
> - Moreover, we have performed further experiments comparing S-TEC with ReLIC for ResNet-50 and we report the results in the revised manuscript. S-TEC with or without a target network outperformed ReLIC as well.
> - To avoid any confusions about the current SOTA, in the revised manuscript we have also included the performance of several recent references into Table 3 that presents our ImageNet results. We have included the SOTA non-contrastive approaches as well, noting also the number of training epochs for each result.
> - We would like to point out that contrastive methods such as ReLIC and its successor ReLICv2 are in fact competitive compared to non-contrastive ones. ReLICv2 is actually the state of the art. On the other hand, ReLIC and ReLICv2 can also be thought of as a hybrid model that uses a contrastive loss and a consistency-based loss similar to the one that drives learning in BYOL (non-contrastive).
> - Our computational resources have not allowed us yet to also perform experiments to compare ReLICv2 (i.e. the SOTA) and the corresponding version of S-TEC.
> - However, we have confirmed in experiments that S-TEC does outperform ReLIC, and ReLIC is the backbone of ReLICv2, i.e., the SOTA.
> - In the revised manuscript we have also evaluated S-TEC and ReLIC in an image segmentation task, which reconfirms S-TEC's improvements compared to a very competitive approach (Table 4).
>
> > Move insights from Appendix to main paper
>
> Our Appendix indeed contained several derivations with theoretical results that have broad implications that were not made clear in the previous main manuscript.
> This undermined the clarity of our work's rigor and potential impact.
> We thank the reviewer for bringing this to our attention.
> In the revised manuscript, we have added a summary of some of these important results, and more explicit pointers to their derivations.
> Moreover, we have reorganized the Results section (section 4) to emphasize also these theoretical results, as opposed to only the experimental results.
>
> In summary, S-TEC's design choices are derived quite rigorously and from first principles of the biological nervous system and of sensory-motor control.
> This point of origin leads to S-TEC as a framework that is rather general (recovers various existing SSL methods, extends them and can potentially generate new ones), biologically-insightful, modular (S-TEC's realization of the manipulation-related inverse model can be kept fixed while its identity-related inverse model can vary), and formal (we have provided detailed derivations of the relation to various SSL methods).
>
> We have now emphasized these key points in the main manuscript.
>
> > Inconsistencies in the Figures.
>
> We have updated the manuscript to correct these errors.

---

> > ### Author Response · Authors · 2022-08-02
> > **Response to Reviewer fHub 2/2**
> >
> > > Show training curves
> >
> > We have included additional Figures in Appendix E.4 (Figure S4-S9) to provide information on the loss progression during training, including manipulation-related losses and identity-related losses, and added text to refer to them in the corresponding results sections (paragraph "Hyperparameter dependence and further details on learning dynamics" in Section 4.2).
> >
> > > 1000 S-TEC epochs vs 200-epoch baselines of AugSelf [1]
> >
> > We agree that these comparisons are important to be performed properly.
> > In order to make the results comparable, we used AugSelf's [1] published code, and we performed optimization using their hyperparameters for 1,000 epochs.
> > We have included these results in Table 2 of our revised manuscript.
> > Conversely, we have trained with our approach also for 200 epochs.
> > S-TEC outperforms AugSelf in the 1000-epoch case, but did not do so in the 200 epoch budget, which we point out in the revised manuscript.
> >
> > > What are possible further improvements?
> >
> > We believe that there are multiple avenues to increase the efficacy of S-TEC in the future.
> >
> > - The positive results that we present have emerged without exploration of several important design choices, which suggests high potential.
> > For example, the number of layers in the projection head after the ResNet is known to be impactful in SSL, and here we have not tuned this for neither the identity-related inverse model nor the manipulation-related inverse model.
> > - Moreover, the improvements that ReLICv2 introduced to ReLIC are certainly compatible with S-TEC offering another possibility.
> > - The specific types of augmentations that we use were merely chosen to match the choice of previous publications, and tuning that choice to S-TEC may improve performance.
> > - In the current tests, the actions that manipulate the objects, generating the augmented inputs, are chosen according to a fixed distribution.
> > Closing the loop between sensory perception and motor action choice may enable the model to choose augmentations that are best for learning better sensory representations.
> >
> > > improve conclusion
> >
> > We have rewritten the conclusion section to address this concern.
> >
> > Altogether, we believe that we have addressed the Reviewer's concerns and we look forward to the updated feedback.
> >
> > ### References
> >
> > [1] Lee, H., Lee, K., Lee, K., Lee, H., & Shin, J. (2021). Improving Transferability of Representations via Augmentation-Aware Self-Supervision. Advances in Neural Information Processing Systems, 34, 17710–17722.

---

> > > ### Comment · Reviewer_fHUb · 2022-08-05
> > > **Happy to improve score**
> > >
> > > The authors have done a great job to address most of the issues, but the main one around novelty remains. I think raising the score to 5 is fair given the paper has been improved and the responses reflect that.

---

> > > > ### Author Response · Authors · 2022-08-08
> > > > **Response to Reviewer fHUb**
> > > >
> > > > We thank the Reviewer for the earlier comments that improved our manuscript, and for acknowledging that most concerns could be addressed, e.g. the results and comparisons that had been raised as the main limitation of our work.
> > > >
> > > > Since novelty had previously not been expressed by the Reviewer as a concern, we would like to emphasize that there are in fact several novel contributions in our work. Specifically, our work has
> > > > - achieved a higher raw experimental score than previously proposed methods could, see ablation studies in Appendix E.2 and E.3,
> > > > - connected SSL methods (not limited to SimCLR) under a common theoretically rigorous framework to neuroscientific principles (see Section 3, 4.1 and Appendix D), and
> > > > - made novel experimentally-testable predictions about the influence of efference copies on sensory learning in the nervous system (see Introduction and Conclusion sections).
> > > >
> > > > These contributions are not only novel, but also important, highly non trivial to arrive at, and interdisciplinary. We have added emphasis to this novelty in the latest revision of the manuscript.
> > > > We hope that, after these clarifications, these aspects can now be taken into account, and we thank the Reviewer again for providing valuable input to our work.

---

### Meta-Review · Area_Chair_xrkW · 2022-08-27

**Recommendation:** Accept
**Confidence:** Less certain

**Metareview:**

This paper introduces a neuroscience-inspired approach for self-supervised learning and makes the connections between SSL and the idea of an Efference Copy (EC) in neuroscience.

The authors provided a number of experiments showing improvements over SimCLR. However, multiple reviewers raised the concern that there are now many non-contrastive objectives for SSL that are sota, and improvements to SimCLR that have not been considered or compared. In the rebuttal phase, the authors did a number of new experiments to compare with other non-contrastive SSL objectives like BYOL, and also ran new evaluations on speech data where they compare with CPC. Overall, many of the key concerns raised by reviewers appear to be addressed by the rebuttal and new experiments provided by the authors.

While there is some noted overlap with previous methods (AugSelf) that diminished some of the enthusiasm, the established connection to neuroscience and efference was still seen as significant even in light of some methodological similarities.


**Award:**

No

---

### Decision · Program_Chairs · 2022-09-14

Accept